# Patients with Inflammatory Bowel Disease Show Fewer Sex-Related Differences in Their Dietary Behavior Than the General Population: A Qualitative Analysis

**DOI:** 10.3390/nu16172954

**Published:** 2024-09-02

**Authors:** Lea Pueschel, Fabian Kockelmann, Momme Kueck, Uwe Tegtbur, Masoumeh Attaran-Bandarabadi, Oliver Bachmann, Heiner Wedemeyer, Henrike Lenzen, Miriam Wiestler

**Affiliations:** 1Department of Gastroenterology, Hepatology, Infectious Diseases and Endocrinology, Hannover Medical School, 30625 Hannover, Germany; pueschel.lea@mh-hannover.de (L.P.); wedemeyer.heiner@mh-hannover.de (H.W.); lenzen.henrike@mh-hannover.de (H.L.); 2Department of Surgery, Hospital Dortmund, University Hospital of the University Witten/Herdecke, 44137 Dortmund, Germany; fabian.kockelmann@uni-wh.de; 3Department for Human Medicine, Witten/Herdecke University, 58455 Witten, Germany; 4Clinic for Rehabilitation and Sports Medicine, Hannover Medical School, 30625 Hannover, Germany; kueck.momme@mh-hannover.de (M.K.); tegtbur.uwe@mh-hannover.de (U.T.); 5Internal Medicine, Private Practice, 30952 Ronnenberg, Germany; 6Department of Internal Medicine I, Siloah St. Trudpert Hospital, 75179 Pforzheim, Germany; obachmann@icloud.com

**Keywords:** dietary behavior, ultra-processed foods and drinks, sex-related differences, inflammatory bowel disease, diet quality

## Abstract

Background: Since diet is generally recognized as an important factor directly modulating the gut microbiome, it is also considered a potential environmental triggering factor for the pathogenesis and onset of inflammatory bowel disease (IBD). While the habitual and sex-related dietary behavior of the general population has been the subject of extensive study and reporting, data on IBD patients’ dietary behavior and especially its sex-related differences are underrepresented. However, as diet is an important factor in the course of IBD, we hypothesized that men and women with IBD have a different dietary profile than the general population. Methods: We performed a cohort analysis of a monocentric, cross-sectional study and compared the sex-related dietary behavior of 82 IBD patients (*n* = 40 women) to a sex- and age-matched cohort of the general German population [*n* = 328 (*n* = 160 women)]. Further on, disease-related quality of life and fecal calprotectin were correlated to the IBD patients’ dietary behavior. Results: While sex-related dietary behavior was frequently of statistical difference in the general population within the IBD cohort, only minor numerical differences were observed between the sexes, which were rarely statistically significant. However, correlation analyses of disease-related quality of life (IBDQ) and diet revealed significant differences in male IBD patients but not in female IBD patients (*p* = 0.007; r = 0.409 for energy intake (kJ/d); *p* = 0.003, r = 0.449 for adherence to Mediterranean diet). Conclusion: The dietary behavior of IBD patients showed more similarity between the sexes than the general German population. Distinct sex-related trends and differences in correlation with disease parameters demonstrated a significant difference for an adaptive dietary behavior, especially in IBD men.

## 1. Introduction

Inflammatory bowel diseases (IBDs), predominantly ulcerative colitis (UC) and Crohn’s disease (CD), are now common in the entire developed world, and incidence rates are rising worldwide [1]. IBD and diet appear to be intrinsically governed by bidirectional interactions: while an active disease can affect one’s nutrient intake, patients’ dietary behavior appears to be a factor that can be either potentially protective or might be contributing to the onset of disease. Recent pilot studies suggest that a westernized diet (e.g., ultra-processed foods) is influencing the risk of developing IBD and exacerbating disease symptoms [2,3,4,5]. Indeed, recent data showed an adverse effect of ultra-processed food on health in general, ranging from obesity to cardiovascular diseases but also inflammatory diseases such as IBD [3,6,7]. IBD research has recently shifted to focusing on identifying dietary patterns and their specific influence on the development and disease course, with popular dietary strategies of IBD patients including avoidant (gluten free, lactose free) and restrictive diets (low FODMAP, CDED) as well as the more balanced “Mediterranean diet” [2,8]. In terms of individualized treatment strategies for IBD, one should emphasize diet as part of a holistic patient-centered approach. However, little attention has been paid to sex-related dietary behavior in the field of IBD, even though we know from the general population that there are significant differences in the dietary behavior of men and women, with different health consequences [9,10]. Therefore, investigating sex-related dietary profiles of IBD patients in direct comparison with a sex- and age-matched cohort of the general population as controls seemed advisable.

### Objectives

The primary objective was to evaluate the sex-related dietary behavior of IBD patients and to assess the presence of distinct nutritional patterns. The secondary aim was the comparison of dietary behavior between IBD patients and the general German population.

## 2. Materials and Methods

We performed a cohort analysis of a monocentric, cross-sectional study with the aim to investigate sex-related dietary behavior of IBD patients compared to a sex- and age-matched cohort of the general German population. Physical activity analysis of the IBD cohort has been reported on previously [11].

### 2.1. Participants and Setting

A total of 94 patients, with a confirmed diagnosis of either ulcerative colitis (UC) or Crohn´s disease (CD), were screened in the IBD outpatient clinic of the Hannover Medical School in the period from January 2017 to October 2017. Of those patients, 91 were recruited into the study. Written informed consent was obtained from each patient prior to screening. Inclusion criteria were a disease duration of at least three months, a confirmed IBD diagnosis of either UC or CD, and disease activity of (a) moderate to severe disease activity or (b) remission. Ostomy, pregnancy, and limitations in everyday life due to cardiovascular or orthopedic diseases were exclusion criteria. The IBD cohort under consideration is ethnically diverse, with the majority of patients identifying as Caucasian.

#### Control Group

The cohort of the German Health Interview and Examination Survey (DEGS1) [12] was chosen as a control group for food consumption, macronutrient intake, and UPFD consumption. The DEGS1 cohort consists of 7987 subjects (*n* = 4198 women); data were collected from 2008 to 2011 and made available by the Robert Koch Institute (RKI) as part of a scientific use agreement [13]. This cohort represents a cross-section of the general German population and comprises ethnically diverse and nationally representative survey and measurement data on adults aged 18 to 79 years living in Germany, with the majority of individuals identifying as Caucasian.

### 2.2. Study Variables and Definitions

#### 2.2.1. Dietary Behavior

In the scope of this study, we understand dietary behavior to mean the composition and quality of daily food intake as well as the determination of daily macronutrient intake. In this work, therefore, dietary behavior must be distinguished from eating habits.

#### 2.2.2. Mediterranean Diet Score

Adherence to a Mediterranean diet was adapted from Trichopoulou et al. [14] based on the sex-specific mean for selected food groups. Food intake was calculated based on the FFQ answers. One point each was awarded whenever the consumption of positive associated foodstuff (vegetables, legumes, fruits and nuts, cereal, fish) was at or above the mean. For negative associated foodstuff (meat, poultry, dairy products), one point each was awarded if the consumption was below the mean. Fat intake was calculated based on the FFQ macronutrient analyses with the ratio of monounsaturated lipids (g) to saturated lipids (g). For ethanol intake, the sex-related values described by Trichopoulou et al. [14] were employed. Failing to reach these thresholds resulted in a score of 0. Overall, the MDS score ranges from 0 to 9, with 9 being maximal adherence to the Mediterranean diet.

#### 2.2.3. Ultra-Processed Foods and Drinks

To calculate ultra-processed food and drink (UPFD) consumption, the NOVA food classification [15] was used to identify mainly Class 4 FFQ items. Tinned/preserved fruits, which are assigned Class 3, were also included due to the frequently added high level of industrial sugars or artificial sweeteners such as cyclamate, as well as additives such as calcium lactate or erythrosine. When calculating UPFD, a distinction was made between foods and drinks. In addition, we calculated the energy content (kJ), the percentage of total energy intake (EN%), and the total weight (g), as so-called “light” or “diet” ultra-processed drinks (UPD) are often consumed in copious quantities but are not high in calories.

#### 2.2.4. Food Groups of the German Nutrition Society

Food groups were created from the FFQ corresponding to the German Nutrition Society (DGE) guideline of food groups [16].

#### 2.2.5. Food Frequency Questionnaire Variables and Macronutrients

The food frequency questionnaire (FFQ) was analyzed by re-coding each question according to the amount and frequency of a single serving. Subsequently, a new variable showing the average daily amount was computed for every question by using the formula for the average daily amount: portion amount × frequency/28 [17,18]. For the subsequent evaluation of the nutrient intake, variables were then calculated using the reference data from the German food nutritional database (Bundeslebensmittelschlüssel [BLS]) [19]. Of the 53 FFQ questions on the frequency of consumption, 52 were converted into variables with the corresponding values of the following nutrients: energy (kJ), protein (g), fat (g), carbohydrates (g), fiber (g), sugar (g), cellulose (mg), lignin (mg), soluble fats (mg), insoluble fats (mg), tyrosine (mg), tryptophan (mg), saturated fats (mg), short-chain fatty acids (mg), medium-chain fatty acids (mg), long-chain fatty acids (mg), omega-3 fatty acids (mg), and omega-6 fatty acids (mg). Energy percentages of the following nutrients were converted using the quantities given here: carbohydrates (17 kJ per gram), proteins (17 kJ per gram), fats (37 kJ per gram), and fiber (8 kJ per gram). To simplify reporting, items were clustered; however, mean comparisons were performed for each item individually.

### 2.3. Data Sources/Measurement

#### 2.3.1. Questionnaires

Patients were asked to complete various questionnaires, with a demographic survey including data on sex, age, body weight, and height. In addition, and depending on disease entity, disease activity was determined via a physician-guided interview including the Harvey Bradshaw Index (HBI) [20] and the Simple Clinical Colitis Activity Index (SCCAI) [21]. To further measure IBD-related quality of life, the German version of the 36-item IBDQ was employed [22,23]. The Food Frequency Questionnaire (FFQ), developed as part of the “German Health Interview and Examination Survey for Adults” (DEGS11) by the RKI [12], was used as a validated collection tool for nutritional data. The questionnaire retrospectively records the frequency and amounts of food consumption in a total of 53 categories for the last four weeks. In addition, questions are asked about eating and cooking habits in relation to the consumption of animal products, fats, and freshly prepared foods. Subsequent nutrient conversion was based on a recently developed tool [19].

#### 2.3.2. Laboratory Values

Biomaterials (blood and fecal samples) were collected as part of the screening visit for biochemical analysis. Parameters included hemoglobin (g/dL), leukocyte counts (1.000/mL), *C*-reactive protein (CRP) (mg/L), vitamin D3 (ng/mL), and calprotectin (mg/kg).

### 2.4. Study Size

A priori sample size estimation for the original study specified the inclusion of ≥50 patients each in “active disease” and “remission” groups. This group allocation was not used for this cohort analysis; instead, patients were divided according to their biological sex. Due to missing FFQ data, *n* = 9 patients were excluded from the cohort analysis.

### 2.5. Statistical Analysis

Statistical analysis was performed with SPSS Statistics software version 28.0.1.0 (SPSS, IBM, Armonk, NY, USA) and GraphPad PRISM version 10.2.3 (GraphPad Software, Boston, MA, USA). Normal distribution was assessed by the Shapiro–Wilk test. Non-normally distributed data are reported as the median (IQR), except for food data, since reporting the median would result in zero values for low-consumption items. Student’s *t*-test was used for comparisons of sex-related within-group and between-group comparisons of food intake, macronutrient intake, UPFD consumption, and adherence to DGE recommendation for the IBD cohort and the DEGS1 cohort. For categorical baseline characteristic variables, the Bonferroni correction was applied to Fisher’s exact test. For metric variables, mean difference was assessed using Student’s *t*-test. For sex-related distribution of diet quality, the Bonferroni correction was applied to Fisher’s exact test. To evaluate sex-related correlations between food data and IBD-related quality of life, we performed precise correlation analyses using Spearman correlation analysis. Food intake, macronutrient intake, UPFD consumption, and mean difference to the DGE reference value are reported as the arithmetic mean, the 95% confidence interval (CI), and the level of significance (*p*). If not stated otherwise, significance levels are two-sided. Clinical relevance is reported as effect size estimate (d).

#### 2.5.1. Missing Data

Participants who did not complete the FFQ were excluded from the analysis. In the case of individual missing data, it could be assumed that these were either missing completely at random (MCAR) or missing at random (MAR). These cases were omitted analysis by analysis.

#### 2.5.2. Sampling Strategy

To avoid distortion when comparing the IBD cohort with the general population of the DEGS1 cohort, matching was carried out. DEGS1 study participants with IBD were identified and filtered (*n* = 102); additionally, all cases were removed in which daily energy intake (EI) could not be calculated due to missing information in the FFQ. A random sample was then drawn which corresponded to the IBD cohort in a ratio of 4:1, and data were sex- and age-matched.

#### 2.5.3. Bias

Misreporting of energy intake in patient-reported outcomes is a well-documented concern [24]. Therefore, misreporting of energy intake was calculated according to Black’s adjustment [25] of Goldberg et al. [26], with the ratio of the reported energy intake (EI_rep_) to the resting energy expenditure (REE) < individual cut-off for underreporting and EI_rep_:REE > 2.4 for overreporting. EI_rep_ was calculated by computing FFQ answers with the reference data from the BLS [19]. While energy expenditure (EE) and total daily energy expenditure (TDEE) were calculated over seven days via the biaxial accelerometer (BodyMedia, Pittsburgh, PA, USA), REE was calculated with the sex-specific Mifflin–St Jeor equations [27], revised from the Harris–Benedict equations [28]. To account for the small study population, the estimated effect size (d) is reported in addition to the statistical significance (*p*), since (d) does not depend on sample size.

## 3. Results

### 3.1. Descriptive Data

Of 94 patients screened, 91 were enrolled and reported on in the original monocentric, cross-sectional study. Due to incomplete FFQ data, *n* = 9 had to be excluded. Consequently, the data from 82 patients were stratified by sex (*n* = 40 women) (Figure 1).

Baseline characteristics were well balanced between the sexes, with no significant differences between demographics, laboratory, and IBD-related variables. Median age for men was 38.8 and for women 36.7. Median BMI for men included in this analysis was 26.1 and thus slightly above the national average of 25.5, while median BMI for women was 24.2 and thus slightly below the national average of 25.2 [29]. Median physical activity level (PAL) for men was 1.7, and median PAL for women was 1.6. (Table 1: Baseline characteristics). We saw a discrepancy between TDEE and EI_rep_ in the IBD cohort. The actual PAL-based energy requirement showed differences to EI_rep_ for both sexes of the IBD cohort. Men in particular showed a higher actual need than could be covered by EI_rep,_ with an observed discrepancy of up to 1838 kJ.

### 3.2. Main Results

#### 3.2.1. IBD Cohort Versus DEGS1 Cohort

##### Macronutrient Intake—Mean Daily Amount

The mean daily intake of macronutrients was calculated from food frequency questionnaires (as described above in Section 2.2.4) and showed no significant differences between the sexes in the IBD patient cohort. However, within the general German population, distinguished sex-related differences have been found in the mean daily intake of proteins (g) (*p* < 0.001, d = 0.7), fat (g) (*p* < 0.001, d = 0.7), and sugar (g) (*p* = 0.006, d = 0.3) and in the energy percentage of carbohydrates (%EN) (*p* = 0.040, d = −0.2) and fiber (%EN) (*p* < 0.001, d = −0.5). Further significant differences were observed for the mean daily macronutrient intake of soluble fats (mg) (*p* = 0.014, d = 0.3), tyrosine (mg) (*p* < 0.001, d = 0.6), tryptophane (mg) (*p* < 0.001, d = 0.7), saturated fats (mg) (*p* < 0.001, d = 0.6), short-chain fatty acids (mg) (*p* = 0.002, d = 0.3), medium-chain fatty acids (mg) (*p* < 0.001, d = 0.5), long-chain fatty acids (mg) (*p* < 0.001, d = 0.7), omega-3 fatty acids (mg) (*p* < 0.001, d = 0.5), and omega-6 fatty acids (mg) (*p* < 0.001, d = 0.8). While no significant differences between IBD men and DEGS1 men were detected, compared to DEGS1 women, IBD women had a higher intake of protein (g) with a mean difference of *p* = 0.04, a higher intake of tryptophan (mg) with a mean difference of *p* = 0.02, and a higher intake of omega-6 fatty acids (mg) with a mean difference of *p* = 0.04. Men and women of both cohorts consumed on average more protein (g) daily (IBD men: 93 g; IBD women: 78 g; DEGS1 men: 95 g; DEGS1 women: 68 g) than the recommended sex-related value of the DGE (men: 57–67 g; women: 48–57 g). The recommended maximum daily fat intake of 30%EN was met by both cohorts (IBD men: 30%; IBD women: 29%; DEGS1 men: 29%; DEGS1 women: 28%), while the recommended fiber intake of ≥30 g daily was not reached by both sexes in the IBD (men: 21 g; women: 18 g) and DEGS1 (men: 21 g; women: 19 g) cohorts (Table 2).

##### Ultra-Processed Foods and Drinks—Mean Daily Amount

Ultra-processed food and drink consumption was calculated as described above (see Section 2.2.3). Minor differences in ultra-processed food and drink consumption between men and women in the IBD cohort were observed. However, those differences were not statistically significant (UPFD *p* = 0.397, d = 0.2; UPD *p* = 0.483, d = 0.2; UPF *p* = 0.262, d = 0.2). In the DEGS1 cohort, medium differences were observed, all of which were statistically significant (UPFD *p* ≤ 0.001, d = 0.5; UPD *p* ≤ 0.001, d = 0.5; UPF *p* ≤ 0.001, d = 0.6). Women of both cohorts on average consumed less ultra-processed foodstuff than their male counterparts, with the highest intake of ultra-processed food (838 g) and ultra-processed drinks (207 g) being reported for healthy men. The lowest intake of UPD was reported for healthy women (366 g), and the lowest intake of UPF was reported for IBD women (146 g) (Table 3).

#### 3.2.2. DGE Food Groups—Mean Daily Consumption

Food groups were created from the FFQ corresponding to the German Nutrition Society (DGE) guideline of food groups [16] (see Section 2.2.4).

##### Fruits and Vegetables

Men in the IBD cohort ate more raw and processed vegetables (65 g) than women within the same cohort (41 g). However, women in the DEGS1 cohort reported a higher intake of raw (69 g) and processed vegetables (43 g) than their male counterparts (56 g and 41 g, respectively). The sexes of both cohorts fell short of the DGE recommendation for fruit and vegetable intake of 550 g daily. The highest mean daily intake of fresh fruits was reported by women in the DEGS1 cohort (238 g), followed by IBD women (193 g). Healthy men reported the lowest intake of fresh fruits (150 g), which showed a significant sex-related difference in comparison (*p* ≤ 0.001; d = −0.4), but they had a higher intake of processed fruits (6.4 g) than their female counterparts (5.4 g). The highest intake of processed and canned fruits was reported by IBD women (9.2 g) and the lowest by IBD men (5.4 g) (Table 4).

##### Juices

The mean daily intake of beverages showed numerical differences between the sexes in the IBD cohort, none of which were significant. IBD men reported a higher intake of vegetable juices (9.9 g; women = 6.7 g; *p* = 0.633, d = 0.1), whereas IBD women had a higher intake of fruit juices (273 g; men = 120 g; *p* = 0.234, d = −0.2). DEGS1 women reported a higher intake of vegetable juices (4.2 g; men = 2.2 g; *p* = 0.145, d = −0.2) and fruit juices (245 g; men = 208 g; *p* = 0.476, d = −0.1). The DGE recommendation of 57.14 g per day was exceeded by both sexes in both cohorts (Table 4).

##### Legumes and Pulses

Men in the IBD cohort ate more raw and processed vegetables (65 g) as well as legumes and pulses (19 g) than women of the same cohort (41 g and 12 g). Women in the DEGS1 cohort reported a mean daily intake of 10 g and men an intake of 12 g. The DGE recommendation for legumes and pulses (a weekly portion size of 125 g computed to 17.85 g daily) was only met by men in the IBD cohort (Table 4).

##### Nuts and Seeds

However, IBD men had the highest daily intake of nuts and seeds (3 g), followed by DEGS1 women (2.1 g); IBD women reported an intake of 1.8 g and DEGS1 men an intake of 2 g. Sexes of both cohorts, however, did not reach the daily amount for nut and seed intake recommended by the DGE of 25 g daily (Table 4).

##### Potato Products

On average, men in the DEGS1 cohort consumed the most fried potatoes (10.1 g) and French fries (13.4 g) daily, followed by men with IBD (7.4 g; 13.3 g). Significant sex-related differences were observed in the DEGS1 cohort for fried potatoes (*p* = 0.013, d = 0.3) and French fries (*p* ≤ 0.001, d = 0.4). Women with IBD reported the highest daily intake of boiled potatoes (70.7 g) and the lowest intake of fried potatoes (6.1 g), with an additional daily intake of 8.8 g of French fries. Women in the DEGS1 cohort had the lowest daily intake of boiled potatoes (58.9 g) and French fries (7.8 g), with an additional mean daily intake of 6.9 g of fried potatoes. Men with IBD consumed on average 62.3 g of boiled potatoes daily, slightly below the intake of healthy men (68.6 g). The DGE recommendation of 250 g weekly (or 35.7 g daily) for potato intake was therefore exceeded by both sexes of both cohorts (Table 4).

##### Butter and Margarine

Mean daily intake of spreadable fats (butter and margarine) showed no significant sex-related differences between IBD men and women as well as no significant differences between the cohorts. IBD men and women reported a higher daily intake of spreadable fats (men = 10.05 g; women = 8.28 g) than their same-sex counterparts in the DEGS1 cohort (men = 9.36 g; women = 6.36 g). The differences between the sexes in the DEGS1 cohort were significant (*p* = 0.003; d = 0.3) (Table 4).

##### Dairy Products

Healthy men reported the highest mean intake of milk (311 g); in comparison with the milk intake of healthy women (213 g), this showed a significant sex-related difference (*p* = 0.0015, d = 0.3). Healthy men further reported the highest intake of yoghurt products (91.5 g), while men with IBD had the highest mean intake of cheese (40.6 g), cream cheese (7.2 g), and eggs (30.6 g). The DGE recommendation for dairy products of 500 g daily was not met by men and women of both cohorts (Table 4).

##### Fish

The DGE recommends a weekly intake of fish (240 g; 34.3 g daily). While IBD men and women had a higher intake of cold (men = 11.9 g; women = 12.4 g) and warm fish (men = 12.2 g; women = 10.7 g) than healthy men and women (cold fish: men = 5.5 g; women = 7.8 g; warm fish: men = 9.4 g; women = 8.8 g), the recommendation was not met by either sex or cohort. However, the DGE guideline points out that the recommendation for weekly portion sizes for fish and meat also includes the other group (Table 4).

##### Meat and Poultry

Men and women with IBD reported an almost identical mean daily intake of poultry (men = 45.5 g; women = 45.6 g). Compared to the DEGS1 cohort, a significant difference for poultry consumption was found between IBD women and DEGS1 women (24.5 g; *p* = 0.030, d = 0.6). The sex-related difference in the DEGS1 cohort was also significant (men = 32.6; *p* = 0.048, d = 0.2). DEGS1 men reported the highest intake of meat (53.8 g; women = 26.5 g; *p* < 0.001, d = 0.6). The DGE guideline recommends a weekly maximum of 240 g (34.3 g daily) for the intake of meat products (including beef, pork, and poultry). The recommended intake was on average exceeded by men and women of both cohorts (Table 4).

##### Cold Cuts

The highest daily intake of cold cuts was reported by DEGS1 men (34 g; women = 14.4; *p* < 0.001, d = 0.7), with a significant difference to IBD men in cold cut intake (19.3 g) (*p* = 0.015, d = −0.4). For women with IBD, a daily intake of 14.5 g was reported. Overall, the recommended intake of 60 g weekly (8.57 g daily) was exceeded (Table 4).

##### Eggs

Men with IBD reported the highest daily intake of eggs (30.56 g), followed by IBD women (26.63 g). DEGS1 men reported an average intake of 18.51 g per day, followed by DEGS1 women (15.48 g). The recommended intake of a weekly portion size of 60 g of eggs (8.57 g daily) was exceeded by all (Table 4).

##### Cereal Products and Rice

Men and women with IBD reported a nearly identical intake of breakfast cereals (men = 3.3 g; women = 3.4 g) but showed a numerical yet not significant difference in the intake of muesli (men = 13 g; women = 6.1 g; *p* = 0.372, d = 0.2). The differences in bread and roll consumption are defined by the flour used, with men and women in the DEGS1 cohort reporting a higher mean daily intake of whole grain bread and rolls (men = 55.7 g; women = 60.5 g); while men with IBD had an intake of 45.1 g, women with IBD reported an intake of 28.4 g, with the difference to healthy women being significant (*p* = 0.001, d = −0.4). The difference in consumption of mixed-flour bread was significant between men with IBD (44.2 g) and healthy men (78.2 g; *p* = 0.008, d = −0.3). DEGS1 women reported the lowest intake (37 g), with the sex-related difference to DEGS1 men being significant (*p* ≤ 0.001, d = 0.5). The IBD cohort reported the highest mean daily intake of white bread and rolls (men = 48.2 g; women = 49.1 g) and the highest intake of rice (men = 31.6 g; women = 22.5 g). The mean daily intake reported by healthy men was the highest for pasta (43.6 g; women = 32.4 g). The DGE recommends consuming a total of 300 g from the food group “cereals, bread, pasta,” and at least one-third of these should be whole grain products. Neither sexes nor both cohorts met this recommendation (Table 4).

### 3.3. IBD: Sex-Related Trends and Differences in Correlation with Disease Parameters

Food uptake of IBD patients showed significant sex-related trends and differences in correlation with disease-related quality of life (IBDQ). For men, IBDQ positively correlated with daily energy intake (EI) (*p* = 0.007; r = 0.409). Furthermore, for men, IBDQ positively correlated with the Mediterranean Diet Score (MDS) (*p* = 0.003, r = 0.449). Fecal calprotectin as an objective disease activity parameter showed no significant correlation with the EI or the MDS, though distinct negative trends in men are demonstrated. Ultra-processed food and drink intake (UPFD) showed no significant sex-related correlation with IBDQ or fecal calprotectin (mg/kg) (Figure 2).

## 4. Discussion

This analysis shows that the dietary behavior of individuals with IBD exhibits fewer sex-related differences compared to the general German population. This observation may be attributed to the unique challenges and dietary restrictions that IBD patients face, regardless of their sex. The majority of IBD patients modify their diets after diagnosis. One study reported that even 75% of patients modified their dietary habits following their diagnosis [30]. This behavior is observed in both men and women with IBD, as the need to manage symptoms and prevent flares becomes a primary concern, superseding sex-specific dietary preferences that might be more pronounced in the general population. IBD patients frequently modify their diets in response to their disease condition. The majority of IBD patients, irrespective of their sex, consider diet to be a significant factor in their disease management [31]. A study found that 85.4% of IBD patients believed that dietary factors could potentially trigger disease relapses, while 32.9% of the same cohort thought that dietary factors could initiate the disease itself [30]. This shared belief system appears to transcend sex boundaries, resulting in comparable dietary behaviors among male and female IBD patients. One of the most prevalent dietary behaviors observed in individuals with IBD is the avoidance of certain foods. A significant proportion of patients, irrespective of sex, impose dietary restrictions in an attempt to prevent disease relapses. For example, 81.7% of patients in one study indicated that they believed it was necessary to eliminate certain products from their diets [30]. The most commonly avoided foods include spicy and fatty foods, raw fruits and vegetables, alcohol, legumes, cruciferous vegetables, and dairy products. This pattern of avoidance is consistent across both male and female patients, indicating that the disease’s impact on dietary choices supersedes typical sex-based differences in food preferences. Interestingly, in our analysis, the consumption of fruits and vegetables did not differ between the IBD patients themselves. However, in comparison to the DEGS1 cohort, female IBD patients especially consumed significantly smaller amounts of fresh vegetables. This may indicate that female IBD patients especially adapt their dietary behavior in this context. When analyzing the general daily energy intake, again no significant differences between IBD patients have been found, whereas in the DEGS1 cohort, men showed a significant higher energy intake compared to women. Moreover, for all macronutrients, we found significant sex-related differences in the general German population, whereas these trends were not significant within the IBD patients. These general sex-related differences in energy uptake and macronutrient consumption are consistent with previously published data. Cross-sectional data from 210.106 men and women of the United Kingdom revealed sex differences in energy intake and distribution. Men had greater intakes of energy and were less likely to have energy intakes above the estimated average requirement. While adherence to recommended dietary guidelines was suboptimal in both sexes, women were significantly more likely than men to exceed recommended intakes of total sugar, total fat, and saturated fat. In contrast, men were more likely to have intakes under the recommended amounts of polyunsaturated fat, carbohydrates, and protein [32]. These differences are partly due to differences in the physiological composition between the sexes [33], but as sex differences in energy and macronutrient intakes vary also by age and socioeconomic status [32], this indicates the necessity for tailored interventions to optimize dietary behavior in men and women across the life course. However, current general dietary recommendations do not incorporate sex-specific factors. The exception to this is during the periods of pregnancy and lactation in women. Despite evidence indicating clear sex-related differences in dietary habits, metabolic processes, hormonal influences, and health outcomes, these factors have not yet been incorporated into dietary recommendations. Dietary recommendations for IBD patients are becoming more and more evident these days. The guidelines from the European Society for Clinical Nutrition and Metabolism (ESPEN) recently recommended a diet rich in fruits and vegetables, rich in *n*-3 fatty acids, and low in *n*-6 fatty acids for adult IBD patients. Furthermore, ultra-processed foods and dietary emulsifiers such as carboxymethylcellulose could be associated with an increased risk of IBD and, therefore, generally, such exclusions can be recommended [34]. A more detailed examination of these specific recommendations within our IBD collective indicates a discernible numerical trend among IBD men with respect to an alteration in dietary behavior when compared to the general population of the DEGS1 cohort. To illustrate, with respect to fruit and vegetable consumption, there is a discernible numerical discrepancy, with a proclivity towards greater intake among IBD men in comparison to men in the DEGS1 cohort. With regard to omega-3 fatty acid consumption, both sexes of the IBD cohort demonstrate a numerically higher intake than that observed in the DEGS cohort. With regard to UPFD consumption, IBD men exhibited numerically significant lower consumption compared to DEGS1 men, while IBD women demonstrated higher UPFD consumption compared to DEGS1 women. Correlation analyses between dietary behavior and disease-related quality of life as measured by the IBDQ also demonstrated that caloric intake and healthy dietary patterns as measured by the Mediterranean Diet Score exhibited significant changes with quality of life, particularly in IBD men. A further correlation also demonstrated this trend in relation to fecal calprotectin as an objective measure of IBD activity; however, this was only observed as a numerical trend without reaching statistical significance. Thus, these findings imply that especially male IBD patients tend to adapt their dietary behavior to their disease activity, since there is clear evidence that disease-related quality of life correlates with IBD activity [11,35]. Per se, this seems in line with the overall adaptive eating behavior of IBD patients during different phases of their disease activity. Several studies so far have underlined frequently restricted foods by IBD patients, e.g., dairy products, processed meats, soft drinks, alcoholic beverages, and fast food [36,37,38,39]. Furthermore, decreased appetite and reduced enjoyment of eating during IBD relapses appear to impair dietary intake in IBD patients [30,31]. However, it is noteworthy that in our analysis this adaptive dietary behavior was more pronounced in IBD men than in IBD women. Overall, these tendencies of IBD patients to adapt their diet to a concept of “orthorexia nervosa” have been published already. Recently, a prospective cohort of 113 IBD patients and 45 controls showed that the risk of orthorexia among IBD patients is high, at 77%, and significantly higher than that observed in a control group [40]. The prevalence rate of orthorexia is on average 6.9% for the general population [41]. Associations between sex and orthorexia nervosa tendency are not clear yet, but already some studies have found that the orthorexia nervosa tendency was significantly higher in men than in women [41,42,43]. Thus, this specific effect of an adaptive dietary behavior, in line with the emerging dietary recommendations for IBD patients, might be more pronounced in male IBD patients, leading to an equalization between the sexes.

## 5. Conclusions

In the present monocentric, cross-sectional study, the dietary behavior of IBD patients displayed a reduced number of sex-related differences when compared to an age- and sex-matched cohort from the general German population (DEGS1 cohort). Our analysis indicates that, in comparison to the distinct sex-related dietary differences observed in the general population, there is a growing convergence in dietary behavior between the sexes in the IBD collective. This phenomenon may be linked to particular dietary recommendations for individuals with IBD. It appears that IBD patients alter their dietary habits in response to the activity of their disease, whereas adherence to recommended dietary guidelines seems suboptimal in both sexes within the general population. Consistent with this, we observed a significant correlation between disease-related factors and the specific dietary behavior; this effect seems to be more pronounced among men with IBD compared to their female counterparts. These findings highlight the importance of providing condition-specific dietary guidance to individuals with IBD, with a particular emphasis on the nutritional challenges that arise from the disease-specific condition of each patient.

### Limitations

We acknowledge that our analysis has limitations. First, as this is a post hoc analysis, the quality of data is limited and the analysis exploratory in nature. Second, the original patient cohort was small and included only IBD patients in a single, tertiary center. To address this, we chose the DEGS1 cohort as a healthy control group for all FFQ-based analyses. Although the FFQ is a validated and extensive tool to collect data on dietary behavior, it is limited in some significant areas: first, the recall period covers 4 weeks, and a time span of this length might introduce a recall bias when completing the questionnaire. In addition, some items which have become staple foods in many households are not included, such as processed and ultra-processed foods, especially of the vegetarian or vegan variety.

## Figures and Tables

**Figure 1 nutrients-16-02954-f001:**
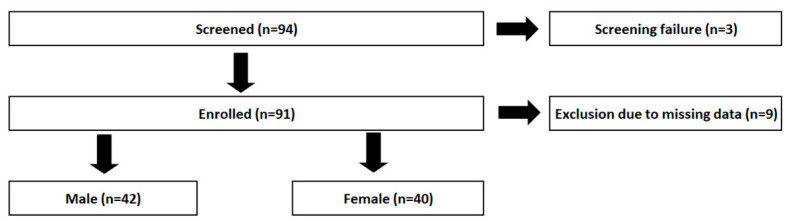
Flowchart of patient enrollment.

**Figure 2 nutrients-16-02954-f002:**
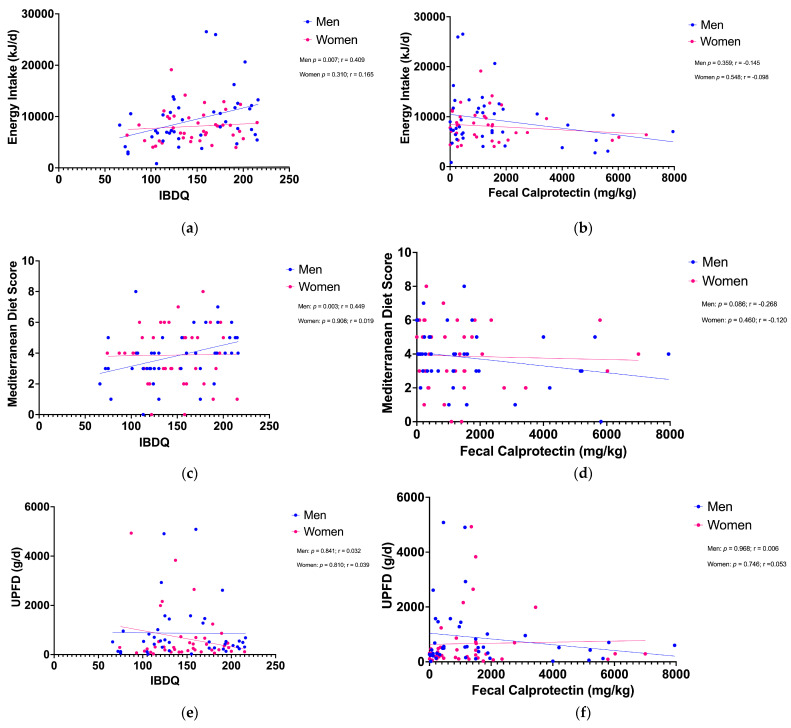
Sex-related trends and differences in correlation with disease parameters: (**a**) Disease-related quality of life positively correlated with daily energy intake (kJ) (*p* = 0.007; r = 0.409) for men but not for women (*p* = 3.310, r = 0.165). (**b**) Biochemical disease activity parameter fecal calprotectin (mg/kg) showed no significant correlation with energy intake (kJ) for men and women. (**c**) IBDQ positively correlated with the Mediterranean Diet Score (*p* = 0.003, r = 0.449) for men but not for women (*p* = 0.908, r = 0.019). (**d**) MDS showed no correlation with fecal calprotectin (mg/kg) for men and women. (**e**) UPFD intake showed no significant sex-related correlation with IBD-related quality of life (**f**) or fecal calprotectin (mg/kg). IBDQ—Inflammatory Bowel Disease Questionnaire; UPFDs—ultra-processed foods and drinks.

**Table 1 nutrients-16-02954-t001:** Baseline characteristics.

		Men (*n* = 42)	Women (*n* = 40)	p_sex_
Entity [*n* (%)]	CD	19 (45.2%)	25 (62.5%)	0.234
	UC	23 (54.8%)	15 (37.5%)	0.233
Disease duration [median (IQR)] (years)		9 [5, 17]	8 [4, 16]	0.781
HBI [mean (SD)]		6.11 [5.92]	7.84 [5.93]	0.393
SCCAI [mean (SD)]		5.7 [5.05]	3.73 [4.68]	0.404
Location of Crohn’s	L1: Ileal	0 (0%)	2 (8%)	0.828
	L2: Colonic	2 (10.5%)	3 (12%)	0.999
	L3: Ileocolonic	12 (63.2%)	12 (48%)	0.999
	L4: Upper GI	5 (26.3%)	8 (32%)	0.999
Crohn’s behavior	B1: Inflammatory	5 (26.3%)	11 (44%)	0.681
	B2: Stricturing	10 (52.6%)	10 (40%)	0.999
	B3: Penetrating	4 (21.1%)	4 (16%)	0.999
UC Montreal classification	E1: Proctitis	2 (8.7%)	1 (6.7%)	0.999
	E2: Left-sided colitis	12 (52.2%)	4 (26.7%)	0.358
	E3: Pancolitis	9 (39.1%)	10 (66.7%)	0.999
BMI [median (IQR)] (kg/m^2^)		25.70 [23.08, 29.08]	23.15 [20.90, 25.60]	0.066
Age [median (IQR)] (years)		36.50 [27.75, 49.25]	33 [24.50, 47.25]	0.492
Smoker [*n* (%)]:	yes	38 (90.5%)	5 (12.5%)	0.999
	no	4 (9.5%)	35 (87.5%)	0.999
Calprotectin [median (IQR)] (mg/L)		1081 [233, 1880]	1020 [269, 1556]	0.588
C-reactive protein [median (IQR)] (mg/L)		2.95 [1.08, 11.03]	2.80 [0.90, 10.48]	0.300
Hemoglobin [median (IQR)] (g/dL)		14.10 [11.90, 14.95]	12.65 [12.10, 13.60]	0.164
Leucocytes [median (IQR)] (Tsd/µL)		8.10 [6.15, 11.20]	8.10 [6.50, 10.93]	0.741
Vitamin D3 25-OH [median (IQR)] (µg/L)		21.83 [13.80, 27.80]	22.22 [13.50, 30.95]	0.870

Values as *n* (%) or median (IQR).

**Table 2 nutrients-16-02954-t002:** Macronutrient intake—mean daily amount.

		IBD Cohort					DEGS1-Cohort					IBD vs DEGS1	
Macronutrients—Mean Daily Amount	Sex	*n*	Mean	95% CI	p_t-Test(twosided)_	*d*	*n*	Mean	95% CI	p_t-Test(twosided)_	*d*	p_t-Test(twosided)_	*d*
Energy (kJ)	Men	42	9358	7658, 11,058	0.205	0.3	168	10,506	9763, 11,249	**<0.001**	0.7	0.184	−0.2
	Women	40	8102	7082, 9122			160	7572	7125, 8019			0.307	0.2
Protein (g)	Men	42	93	72.90, 113.0	0.181	0.3	168	95	87.83, 101.7	**<0.001**	0.7	0.864	0.0
	Women	40	78	67.89, 87.95			160	68	64.31, 72.31			**0.044**	0.4
Protein (%EN)	Men	42	16	15.43, 17.48	0.689	−0.1	168	16	15.15, 16.14	0.731	0.0	0.154	0.2
	Women	40	17	15.46, 18.11			160	16	15.22, 16.33			0.121	0.3
Fat (g)	Men	42	78	59.95, 95.32	0.169	0.3	168	81	74.98, 87.19	**<0.001**	0.7	0.649	−0.1
	Women	40	64	54.30, 73.32			160	57	52.66, 60.77			0.135	0.3
Fat (%EN)	Men	42	30	27.65, 31.72	0.764	0.1	168	29	27.71, 29.74	0.214	0.1	0.400	0.1
	Women	40	29	27.07, 31.42			160	28	26.72, 28.87			0.234	0.2
Carbohydrates (g)	Men	42	272	224.8, 318.4	0.466	0.2	168	323	296.1, 349.9	<0.001	0.5	0.084	−0.3
	Women	40	250	212.8, 286.8			160	242	224.2, 259.1			0.682	0.1
Carbohydrates (%EN)	Men	42	50	47.73, 53.25	0.497	−0.2	168	52	50.17, 52.96	**0.040**	−0.2	0.494	−0.1
	Women	40	52	48.80, 54.97			160	54	52.23, 55.04			0.281	−0.2
Fiber (g)	Men	42	21	15.70, 26.04	0.361	0.2	168	21	19.51, 23.08	0.093	0.2	0.847	0.0
	Women	40	18	15.14, 21.15			160	19	17.61, 20.85			0.546	−0.1
Fiber (%EN)	Men	42	2	1.551, 2.098	0.708	−0.1	168	2	1.590, 1.817	**<0.001**	−0.5	0.367	0.2
	Women	40	2	1.635, 2.155			160	2	1.943, 2.190			0.224	−0.2
Sugar (g)	Men	42	127	98.83, 155.6	0.883	0.0	168	159	140.1, 178.5	**0.006**	0.3	0.123	−0.3
	Women	40	130	97.47, 163.3			160	126	112.8, 139.6			0.790	0.0
Cellulose (mg)	Men	42	3631	2671, 4591	0.609	0.1	168	3056	2802, 3310	0.124	−0.2	0.249	0.3
	Women	40	3322	2587, 4057			160	3367	3057, 3676			0.901	0.0
Lignin (mg)	Men	42	875	604.4, 1145	0.354	0.2	168	724	647.6, 801.0	0.268	−0.1	0.286	0.3
	Women	40	733	589.1, 877.4			160	794	696.2, 891.2			0.566	−0.1
Soluble fats (mg)	Men	42	6401	4799, 8003	0.308	0.2	168	6803	6247, 7359	**0.014**	0.3	0.559	−0.1
	Women	40	5479	4632, 6326			160	5868	5372, 6364			0.476	−0.1
Insoluble fats (mg)	Men	42	14,189	10,654, 17,724	0.386	0.2	168	14,426	13,225, 15,627	0.138	0.2	0.899	0.0
	Women	40	12,390	10,188, 14,592			160	13,185	12,064, 14,306			0.529	−0.1
Tyrosine (mg)	Men	42	3408	2665, 4152	0.206	0.3	168	3482	3208, 3757	**<0.001**	0.6	0.825	0.0
	Women	40	2871	2458, 3283			160	2535	2370, 2699			0.087	0.3
Tryptophan (mg)	Men	42	1098	861.5, 1335	0.193	0.3	168	1115	1032, 1198	**<0.001**	0.7	0.894	0.0
	Women	40	924	800.7, 1048			160	795	748.5, 842.5			**0.024**	0.4
Saturated fats (mg)	Men	42	35,011	27,309, 42,713	0.231	0.3	168	37,108	34,092, 40,123	**<0.001**	0.6	0.561	−0.1
	Women	40	29,386	24,128, 34,643			160	26,320	24,183, 28,457			0.226	0.2
Short-chain fatty acids (mg)	Men	42	1586	1248, 1924	0.356	0.2	168	1607	1428, 1786	**0.002**	0.3	0.915	0.0
	Women	40	1373	1060, 1687			160	1244	1105, 1383			0.422	0.1
Medium-chain fatty acids (mg)	Men	42	1488	1170, 1805	0.419	0.2	168	1544	1378, 1709	**<0.001**	0.5	0.762	−0.1
	Women	40	1307	989.7, 1624			160	1101	999.6, 1203			0.219	0.3
Long-chain fatty acids (mg)	Men	42	67,875	52,087, 83,664	0.148	0.3	168	71,743	66,309, 77,177	**<0.001**	0.7	0.567	−0.1
	Women	40	54,917	46,519, 63,316			160	49,630	46,035, 53,224			0.208	0.2
Omega-3 fatty acids (mg)	Men	42	1955	1287, 2622	0.513	0.1	168	1792	1658, 1927	**<0.001**	0.5	0.633	0.1
	Women	40	1701	1320, 2082			160	1380	1236, 1524			0.065	0.3
Omega-6 fatty acids (mg)	Men	42	10,755	8355, 13,155	0.134	0.3	168	11,047	10,265, 11,829	**<0.001**	0.8	0.816	−0.1
	Women	40	8742	7551, 9933			160	7550	7043, 8056			**0.045**	0.4

Units of daily intake of macronutrients are reported as kilojoules (kJ), grams (g), or milligrams (mg). Results of Student’s *t*-test between the sexes and the IBD and DEGS1 cohorts are given as the arithmetic mean, the 95% confidence interval (CI), the level of significance (*p*), and the estimated effect size (d). P*_t_*_-test_ is printed bold when significant. IBD—inflammatory bowel disease; DEGS1—cohort of the German Health Interview and Examination Survey.

**Table 3 nutrients-16-02954-t003:** Ultra-processed foods and drinks—mean daily amount.

		IBD Cohort					DEGS1-Cohort					IBD vs DEGS1	
Ultra-Processed Foods and Drinks—Mean Daily Amount	Sex	*n*	Mean	95% CI	p_t-Test(twosided)_	*d*	*n*	Mean	95% CI	p_t-Test(twosided)_	*d*	p_t-Test(twosided)_	*d*
UPFD (g)	Men	41	871	510.7, 1232	0.397	0.2	150	968	792.7, 1143	**<0.001**	0.5	0.619	−0.1
	Women	40	664	328.9, 999.0			143	522	389.9, 654.0			0.356	0.2
UPFD (kJ)	Men	41	3179	2421, 3938	0.170	0.3	148	3884	3452, 4317	**<0.001**	0.6	0.127	−0.3
	Women	40	2514	1914, 3115			143	2447	2166, 2728			0.830	0.0
UPD (g)	Men	42	685	338.1, 1032	0.483	0.2	156	838	654.7, 1020	**<0.001**	0.5	0.445	−0.1
	Women	40	518	191.4, 845.4			152	366	244.2, 487.3			0.381	0.2
UPD (kJ)	Men	42	987	535.0, 1440	0.233	0.3	156	1266	1008, 1523	**<0.001**	0.6	0.316	−0.3
	Women	40	647	301.5, 991.7			152	504	377.8, 629.4			0.436	0.2
UPF (g)	Men	41	172	131.8, 212.9	0.262	0.3	158	207	187.9, 226.7	**<0.001**	0.6	0.111	−0.3
	Women	40	146	120.3, 170.8			150	148	134.1, 162.0			0.867	0.0
UPF (kJ)	Men	41	2172	1614, 2729	0.37	0.2	156	2683	2416, 2950	**<0.001**	0.5	0.089	−0.3
	Women	40	1867	1479, 2256			150	1924	1709, 2139			0.404	0.0

Units of daily intake of ultra-processed foods and drinks (UPFDs), ultra-processed foods (UPFs), and ultra-processed drinks (UPDs) are reported as kilojoules (kJ) and grams (g). Results of Student’s *t*-test between the sexes and the IBD and DEGS1 cohorts are given as the arithmetic mean, the 95% confidence interval (CI), the level of significance (*p*), and the estimated effect size (d). P*_t_*_-test_ is printed bold when significant.

**Table 4 nutrients-16-02954-t004:** DGE food groups—mean daily consumption.

			IBD Cohort					DEGS1-Cohort					IBD vs DEGS1	
DGE Food Groups (g/d)		Sex	*n*	Mean	95% CI	p_t-Test(twosided)_	*d*	*n*	Mean	95% CI	p_t-Test(twosided)_	*d*	p_t-Test(twosided)_	*d*
**Fruits and vegetables**	Fresh Vegetables	Men	42	65.29	27.41, 103.2	0.224	0.3	168	56.47	44.85, 68.09	**0.004**	−0.3	0.656	0.1
		Women	40	40.92	28.01, 53.82			159	81.95	68.90, 95.01			**<0.001**	−0.5
	Processed Vegetables	Men	42	68.26	43.70, 92.81	0.91	0.0	167	40.58	32.93, 48.23	0.053	−0.2	**0.035**	0.5
		Women	40	66.45	45.64, 87.25			159	53.15	42.81, 63.48			0.127	0.2
	Fresh Fruits	Men	42	168.08	104.2, 231.9	0.683	−0.1	167	150.25	121.9, 178.6	**<0.001**	−0.4	0.586	0.1
		Women	40	193.09	86.13, 300.0			160	237.61	197.0, 278.2			0.363	−0.2
	Processed Fruits	Men	42	5.36	1.296, 9.418	0.278	−0.2	166	6.39	3.757, 9.022	0.536	0.1	0.716	−0.1
		Women	40	9.16	3.346, 14.97			160	5.43	3.921, 6.935			0.216	0.3
**Juices**	Fruit Juice	Men	42	120.96	69.07, 172.9	0.234	−0.3	167	208.25	150.8, 265.7	0.476	−0.1	0.144	−0.3
		Women	40	273.53	23.54, 523.5			159	245.74	158.2, 333.3			0.797	0.0
	Vegetable Juice	Men	42	9.86	−0.4242, 20.15	0.633	0.1	167	2.22	1.000, 3.448	0.145	−0.2	0.144	0.5
		Women	39	6.68	−1.748, 15.12			159	4.2	1.832, 6.569			0.435	0.1
**Legumes and pulses**		Men	42	19.45	6.867, 32.04	0.272	0.2	168	15.99	11.59, 20.39	**0.024**	0.2	0.525	0.1
		Women	39	11.62	5.613, 17.64			159	10.28	7.993, 12.58			0.314	0.1
**Nuts and seeds**		Men	42	2.9	1.473, 4.336	0.254	0.3	167	1.99	1.206, 2.767	0.842	0.0	0.290	0.2
		Women	40	1.79	0.4449, 3.126			159	2.09	1.377, 2.811			0.699	−0.1
**Potatoes**	Boiled Potatoes	Men	42	62.31	39.31, 85.32	0.596	−0.1	167	68.63	57.33, 79.93	0.167	0.2	0.621	−0.1
		Women	40	70.74	48.51, 92.98			160	58.86	50.69, 67.03			0.316	0.2
	Fried Potatoes	Men	42	7.35	4.979, 9.721	0.475	0.2	168	10.08	8.114, 12.06	**0.013**	0.3	0.079	−0.2
		Women	40	6.08	3.372, 8.782			159	6.86	5.267, 8.463			0.653	−0.1
	French fries	Men	42	13.27	5.886, 20.64	0.292	0.2	166	13.44	11.19, 15.69	**<0.001**	0.4	0.953	0.0
		Women	40	8.81	4.852, 12.76			160	7.79	6.192, 9.385			0.591	0.1
**Butter and margarine**		Men	42	10.05	6.19, 13.91	0.492	0.2	168	9.36	7.77, 10.96	**0.003**	0.3	0.715	0.1
		Women	40	8.28	4.84, 11.72			160	6.36	5.18, 7.53			0.190	0.2
**Dairy products**	Milk	Men	41	247.17	157.5, 336.8	0.723	0.1	165	311.1	243.7, 378.5	**0.015**	0.3	0.376	−0.2
		Women	40	225.18	138.3, 312.0			158	213.33	172.7, 253.9			0.798	0.0
	Cream Cheese	Men	42	7.23	3.345, 11.12	0.950	0.0	167	5.61	4.202, 7.017	0.635	−0.1	0.433	0.2
		Women	40	7.07	3.485, 10.65			160	6.15	4.377, 7.928			0.648	0.1
	Cheese	Men	42	40.65	24.16, 57.14	0.311	0.2	165	33.67	27.53, 39.80	0.882	0.0	0.346	0.2
		Women	40	30.5	19.30, 41.70			160	32.92	24.97, 40.86			0.777	−0.1
	Yoghurt	Men	42	78.53	39.63, 117.4	0.702	−0.1	164	91.52	68.47, 114.6	0.230	0.1	0.605	−0.1
		Women	40	90.49	40.40, 140.6			159	75.57	63.16, 87.99			0.562	0.1
**Fish**	Cold fish	Men	42	30.56	15.96, 45.16	0.692	0.1	168	18.51	15.32, 21.70	0.171	0.2	0.111	0.4
		Women	40	26.63	12.96, 40.29			160	15.48	12.51, 18.44			0.114	0.4
	Warm fish	Men	42	12.2	6.782, 17.52	0.675	0.1	167	9.4	7.605, 11.15	0.32	0.1	0.327	0.2
		Women	40	10.7	6.594, 14.86			160	8.8	7.316, 10.27			0.287	0.2
**Meat and poultry**	Poultry	Men	42	45.5	27.14, 63.87	0.996	0.0	168	32.6	26.02, 39.11	**0.048**	0.2	0.187	0.3
		Women	40	45.6	27.20, 63.94			160	24.5	19.80, 29.16			**0.030**	0.6
	Meat	Men	42	51.8	31.16, 72.51	0.104	0.4	167	53.8	43.86, 63.72	**<0.001**	0.6	0.863	0.0
		Women	40	32.2	20.35, 44.04			160	26.5	22.83, 30.26			0.363	0.2
**Cold cuts**		Men	42	19.3	10.91, 27.73	0.335	0.2	168	34.0	28.50, 39.54	**<0.001**	0.7	**0.015**	−0.4
		Women	40	14.5	8.994, 19.93			160	14.4	11.55, 17.22			0.982	0.0
**Eggs**		Men	42	30.56	15.96, 45.16	0.692	0.1	168	18.51	15.32, 21.70	0.171	0.2	0.111	0.4
		Women	40	26.63	12.96, 40.29			160	15.48	12.51, 18.44			0.114	0.4
**Cereal products**	Breakfast cereals	Men	42	3.3	0.7905, 5.766	0.959	0.0	168	2.9	1.994, 3.881	0.063	0.2	0.765	0.1
		Women	40	3.4	−0.8803, 7.684			159	1.8	0.9100, 2.585			0.448	0.2
	Muesli	Men	42	13.0	−1.690, 27.58	0.372	0.2	166	5.7	3.731, 7.626	0.248	0.1	0.326	0.3
		Women	40	6.1	2.605, 9.605			158	4.3	2.798, 5.693			0.274	0.2
	Wholegrain bread and rolls	Men	42	45.1	12.03, 78.09	0.357	0.2	157	55.7	43.15, 68.23	0.611	−0.1	0.481	−0.1
		Women	40	28.4	15.03, 41.80			160	60.5	46.90, 74.00			**0.001**	−0.4
	Mixed bread	Men	42	44.2	25.83, 62.52	0.897	0.0	166	78.2	60.72, 95.66	**<0.001**	0.5	**0.008**	−0.3
		Women	40	42.6	25.57, 59.56			159	37.0	29.30, 44.65			0.526	0.1
	White bread and rolls	Men	42	48.2	32.88, 63.55	0.955	0.0	166	45.6	37.52, 55.77	**0.006**	0.3	0.875	0.0
		Women	39	49.1	20.11, 78.10			160	30.9	24.19, 37.57			0.222	0.3
	Pasta	Men	42	38.7	20.70, 56.74	0.589	0.1	167	43.6	36.26, 50.84	**0.013**	0.3	0.575	−0.1
		Women	40	33.2	23.79, 42.56			160	32.4	27.48, 37.32			0.888	0.0
	Rice	Men	42	31.6	14.16, 49.04	0.357	0.2	168	18.9	14.77, 22.94	0.759	0.0	0.158	0.4
		Women	40	22.5	13.42, 31.51			160	19.7	16.54, 22.78			0.468	0.1

Food frequency questionnaire (FFQ) variables are sorted by the German Nutrition Society (DGE) guideline of food groups. All units are given as grams per day (g/d). Results of Student’s *t*-test between the sexes and the IBD and DEGS1 cohorts are given as the arithmetic mean, the 95% confidence interval (CI), the level of significance (*p*), and the estimated effect size (d). P*_t_*_-test_ is printed bold when significant. IBD—inflammatory bowel disease; DEGS1—cohort of the German Health Interview and Examination Survey.

## Data Availability

The original contributions presented in the study are included in the article, further inquiries can be directed to the corresponding author.

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
