# Peer review of "Patients with Inflammatory Bowel Disease Show Fewer Sex-Related Differences in Their Dietary Behavior Than the General Population: A Qualitative Analysis"

_nutrients, 2024, doi:10.3390/nu16172954_

Round 1

Reviewer 1 Report

Comments and Suggestions for Authors

The paper has a very interesting topic and the original part is indeed the fact that they studied gender-based differences regarding dietary behaviour. 82 patients were finally analysed, recruited in about 10 months. I want to congratulate the authors for quantifying and analysing in a very meticulous way the diet (taking in consideration also potential sources of bias).

I appreciate the flowchart detailing the recruitment process. However, I think that the "screening failure" arrow should be placed one level above. Please review and if necessary correct.

In think the outcome of the study was quite predictable. Most of the physicians make gender-neutral dietary recommendation in IBD patients. It would have been interesting to see an evaluation of the diet in the moment of diagnosis.

It is difficult to understand the last statement in the Conclusions statement. I know that due sex-specific food preferences, IBD dietary recommendations may have a different impact on males by comparison to females. Also, as the study revealed, dietary recommendation adherence and behaviour may differ based on gender. Both these ideas were a result of the study. However, no discussion regarding existing conventional sex-specific dietary recommendations was included in teh paper. Please review the Conclusions and keep only those generated by the study. Revise and if necessary include specific discussions in this regard.

Lines 514-516: authors state "Our analysis indicates that compared to the established sex-related dietary differences observed in the general population, these differences are becoming increasingly similar in the IBD collective”. I believe what was meant was that the diet behaviour was becoming increasingly similar (or the differences become minimal). Please review and if necessary correct.

Author Response

We would like to express our sincerest gratitude to the reviewers for their invaluable feedback. In light of the individual comments, the initial manuscript was further adapted and corrected.  Please find the detailed responses below and the corresponding revisions/corrections highlighted/in track changes in the re-submitted files.

Reviewer 1, Comments:
1)    I appreciate the flowchart detailing the recruitment process. However, I think that the "screening failure" arrow should be placed one level above. Please review and if necessary correct.
Response: Figure 1 has been adapted and corrected in accordance with the suggestions outlined above.

2)    It is difficult to understand the last statement in the Conclusions statement. I know that due sex-specific food preferences, IBD dietary recommendations may have a different impact on males by comparison to females. Also, as the study revealed, dietary recommendation adherence and behaviour may differ based on gender. Both these ideas were a result of the study. However, no discussion regarding existing conventional sex-specific dietary recommendations was included in the paper. Please review the Conclusions and keep only those generated by the study. Revise and if necessary include specific discussions in this regard.
Response: The final conclusion has been modified in accordance with this thoughtful suggestion. In particular, the previous commentary on conventional sex-specific dietary recommendations has been removed, given that this specific topic was not subjected to detailed examination. Additionally, it is evident that, at present, sex-specific recommendations for the general population encompass only a few minor sex-based distinctions (e.g. caloric intake, pregnancy and breastfeeding). The overarching tenets of a healthy diet remain largely consistent for men and women. Consequently, further research is required to ascertain the potential benefits of more tailored, sex-specific dietary guidelines. However, this question remains unanswered for the general population.

3)    Lines 514-516: authors state "Our analysis indicates that compared to the established sex-related dietary differences observed in the general population, these differences are becoming increasingly similar in the IBD collective”. I believe what was meant was that the diet behaviour was becoming increasingly similar (or the differences become minimal). Please review and if necessary correct.
Response: The statement has been modified in accordance with this thoughtful suggestion. In particular, the explicit wording has been adapted to demonstrate that there is a growing convergence in dietary behaviour between the sexes in the IBD collective.

Again, we would like to extend our gratitude to the reviewers for dedicating their valuable time to a critical reading and correction of this manuscript.

Reviewer 2 Report

Comments and Suggestions for Authors

1Overall this is a reasonable  manuscript, from a scientific perspective, in terms of the goals/aspiration of the study undertaken/reported. Moreover, it is timely and necessary in view of the, unfortunate, widespread occurrence of IBD. The more we know about this condition, however small in a relative context, the better. 

   Minor corrections, to the manucript, include the following. 

11.       Line 20; is the term “course” appropriate?

22.       Line 21; ditto in relation to: “in extension”? “by extension”, should be used instead.

33.       Line 34; what is the meaning of the term “German population”? This term should be clarified in the general text of the manuscript. Do the authors mean “white ethnic Germans”. In addition would it not be worth making clear, again in the general text and not the abstract as to the ethnicity of the cohort of patients examined or do the authors believe that this would not be worth undertaking?

44.       In the Keywords why are some capitalized and others not (i.e., uniformity)?

55.       Is an up-to-date reference at the end of the first sentence of the “Introduction” not required?

66.       Line 64; the authors state that one of their objectives is to examine the nutritional quality of the diet related to the cohort of people examined. Is this really true? If so where is the data and has this objective been met in detail from a scientific perspective? Are they really examining “nutritional quality” or different kinds of foods?

76.       Line 95; replace “paper” by the term “study”.

87.       Line 108; why is the e in ethanol capitalized?

98.       Line 126; “German society for nutrition” should read “German Nutrition Society”.

19.   Line 168: the name for the protein Calprotectin should not be capitalized.

110.   Line 224; full-stop is missing from the end of the sentence and Figure 1 needs to be made more clear and “tidied-up”.

111.   Lines 423/424; the term “energy” should not be capitalized.

Comments on the Quality of English Language

As stated above.

Author Response

We would like to express our sincerest gratitude to the reviewers for their invaluable feedback. In light of the individual comments, the initial manuscript was further adapted and corrected.  Please find the detailed responses below and the corresponding revisions/corrections highlighted/in track changes in the re-submitted files.

Reviewer 2, Comments:

1)    Line 20; is the term “course” appropriate?
Response: As indicated above the term "course" has been replaced with "onset" to better align with the growing evidence that diet is an environmental risk factor in the pathogenesis and onset of inflammatory bowel diseases.

2)    Line 21; ditto in relation to: “in extension”? “by extension”, should be used instead.
Response: The aforementioned error has been corrected through the implementation of additional modifications to the sentence structure.

3)    Line 34; what is the meaning of the term “German population”? This term should be clarified in the general text of the manuscript. Do the authors mean “white ethnic Germans”. In addition would it not be worth making clear, again in the general text and not the abstract as to the ethnicity of the cohort of patients examined or do the authors believe that this would not be worth undertaking?
Response: We are grateful for the reviewers´ insightful contribution to this discussion. In response to this comment, the ethnic structure of the DEGS1 cohort, as well as that of the IBD cohort, has been described in greater detail in the section entitled "Material and Methods." The two cohorts are predominantly Caucasian in ethnicity.

4)    In the Keywords why are some capitalized and others not (i.e., uniformity)?
Response: The aforementioned error has been corrected. In order to ensure uniformity, all keywords have been capitalised.

5)    Is an up-to-date reference at the end of the first sentence of the “Introduction” not required?
Response: A recent reference has been added at the end of the first sentence of the “Introduction”, accordingly.

6)    Line 64; the authors state that one of their objectives is to examine the nutritional quality of the diet related to the cohort of people examined. Is this really true? If so where is the data and has this objective been met in detail from a scientific perspective? Are they really examining “nutritional quality” or different kinds of foods?
Response: We are grateful for the reviewers´ insightful contribution to this discussion. In response to this comment, the term "nutritional quality" has been modified to "distinct nutritional patterns". The Mediterranean diet score is a validated measure of diet quality, as it assesses adherence to the Mediterranean dietary pattern, which is linked to a range of health benefits, particularly in relation to cardiometabolic health. It could therefore be argued that the assessment of diet quality is applicable as we evaluate the Mediterranean diet score within our IBD collective. However, as we also assess consumption of ultra-processed foods and beverages, we have modified the terminology to "distinct nutritional patterns."

7)    Line 95; replace “paper” by the term “study”.
Response: The aforementioned error has been corrected. The term “paper” has been changed to “study”.

8)    Line 108; why is the e in ethanol capitalized?
Response: The aforementioned error has been corrected. The letter "e" in the word "ethanol" has been changed to lowercase.

9)    Line 126; “German society for nutrition” should read “German Nutrition Society”.
Response: The aforementioned error has been corrected. The society's name has been altered to "German Nutrition Society" throughout the text.

10)    Line 168: the name for the protein Calprotectin should not be capitalized.
Response: The aforementioned error has been corrected. The letter "c" in the word "calprotectin" has been changed to lowercase throughout the text.

11)    Line 224; full-stop is missing from the end of the sentence and Figure 1 needs to be made more clear and “tidied-up”.
Response: The aforementioned error has been corrected. A full stop has been added at the end of the sentence. The requisite amendments have been made to Figure 1.

12)    Lines 423/424; the term “energy” should not be capitalized.
Response: The aforementioned error has been corrected. The letter "e" in the word "energy" has been changed to lowercase throughout the text.

Again, we would like to extend our gratitude to the reviewers for dedicating their valuable time to a critical reading and correction of this manuscript.